# Glucagon Control on Food Intake and Energy Balance

**DOI:** 10.3390/ijms20163905

**Published:** 2019-08-11

**Authors:** Omar Al-Massadi, Johan Fernø, Carlos Diéguez, Ruben Nogueiras, Mar Quiñones

**Affiliations:** 1Inserm UMR-S1270, Sorbonne Université, Faculté des Sciences et d’Ingénierie, Institut du Fer a Moulin, 75005 Paris, France; 2Hormone Laboratory, Haukeland University Hospital, N-5020 Bergen, Norway; 3Department of Physiology, CIMUS, University of Santiago de Compostela-Instituto de Investigación Sanitaria, 15782 Santiago de Compostela, Spain; 4CIBER Fisiopatología de la Obesidad y Nutrición (CIBERobn), 15706 Santiago de Compostela, Spain

**Keywords:** glucagon, energy balance, food intake, body weight, energy expenditure, thermogenesis, lipid metabolism, obesity

## Abstract

Glucagon exerts pleiotropic actions on energy balance and has emerged as an attractive target for the treatment of diabetes and obesity in the last few years. Glucagon reduces body weight and adiposity by suppression of appetite and by modulation of lipid metabolism. Moreover, this hormone promotes weight loss by activation of energy expenditure and thermogenesis. In this review, we cover these metabolic actions elicited by glucagon beyond its canonical regulation of glucose metabolism. In addition, we discuss recent developments of therapeutic approaches in the treatment of obesity and diabetes by dual- and tri-agonist molecules based on combinations of glucagon with other peptides. New strategies using these unimolecular polyagonists targeting the glucagon receptor (GCGR), have become successful approaches to evaluate the multifaceted nature of glucagon signaling in energy balance and metabolic syndrome.

## 1. Introduction

### 1.1. Background

In the early 1920s, Kimball and Murlin [1] reported that extracts of pancreatic tissue produced a hyperglycemic response, due to a circulating factor that was identified as glucagon. This pancreatic hormone is a 29-amino-acid peptide released from the α-cells of the islet of Langerhans, and has for long been recognized as the principal counter-regulatory hormone to insulin in response to low glucose [2,3]. Glucagon and glucagon-like peptides are transcribed from a common proglucagon gene, a peptide precursor of 160 amino acids, that is expressed in pancreatic islet α-cells, in a specific population of enteroendocrine cells (L-cells) of the intestinal mucosa, and in a set of neurons in the nucleus tractus solitarius (NTS) of the medulla oblongata [4,5].

### 1.2. Glucagon Peptides

The posttranslational regulation of proglucagon gene generates a number of well-known hormones, such as glucagon and glucagon like peptide 1 (GLP-1), and also other bioactive peptides like glucagon like peptide 2 (GLP-2), glicentin, glicentin-related pancreatic polypeptide (GRPP), and oxyntomodulin (OXM). Most of these proglucagon-derived peptides show very specific effects on glucose and energy balance, although some of them exhibit unclear functional roles [6,7,8,9,10]. The tissues-specific expression of various products from the proglucagon gene depend on posttranslational modifications by specific prohormone convertases (PC) [11]. It is important to note that glucagon in the α-cells is cleaved from the proglucagon gene by the prohormone convertase 2 (PC2), encoded by the Proprotein Convertase Subtilisin/Kexin Type 2 gene [12]. GLP-1, GLP-2, GRPP, and OXM, on the other hand, are cleaved from the proglucagon gene in the brain and intestine through the prohormone convertase 1 (PC1), also known as prohormone convertase 3 and usually abbreviated as PC1/3, encoded by the Proprotein Convertase Subtilisin/Kexin Type 1 gene [13]. 

### 1.3. Glucagon Receptor

Glucagon acts through the binding and activation of glucagon receptor (GCGR). The GCGR belongs to class B of G protein-coupled receptors (GPCRs) located on the cell surface [14]. GCGR is mainly present in liver and kidney and to a lesser extent in intestinal smooth muscle, brain, adipose tissue, the adrenal gland, heart, and in both α- and β-pancreatic cells [15,16]. The GCGR has been considered an important drug target in the treatment of type 2 diabetes mellitus (T2DM) due to its effect on pancreatic alpha-cells. However, additional and novel effects for glucagon, such as modulation of satiety, thermogenesis, energy expenditure, and control of lipid metabolism have more recently been garnering scientific attention [3,17]. 

### 1.4. Objectives

In this review article, we highlight the specific metabolic actions exerted by glucagon that participates in the control of food intake and energy balance. In addition, we discuss the novel therapeutic approaches in the treatment of obesity and diabetes by dual- and tri-agonist molecules based on glucagon in combination with other peptides. 

## 2. The Effect of Glucagon on Satiety and Appetite Suppression 

Glucagon exerts effects on metabolism beyond the regulation of glucose metabolism, including modulation of satiety as demonstrated both in humans [18,19] and in rats [20,21]. The effect of glucagon on satiety was shown to be mainly due to the inhibition of meal size via the liver–brain axis [22]. Concordantly, intraperitoneal injections of antibodies against glucagon caused an increase in meal size in rats [23]. Moreover, it has also been shown that a considerable amount of glucagon is released during meals. These facts support the idea of a physiological role of glucagon in the termination of meals or postprandial satiety [24,25]. The anorexigenic action of glucagon is initiated in the liver, which is able to sense the glucagon levels in the hepatic portal vein and send the information through the vagal afferents to the central nervous system (CNS) (Figure 1). 

Thus, the hepatic branch of the vagus nerve conveys the satiety signal to the area postrema (AP) and the NTS, and from there the neuronal signal is transported to the hypothalamus. The existence of this neuronal loop is supported by experiments in rats, where glucagon administration into the portal vein was unable to induce satiety after hepatic vagotomy [22]. Moreover, specific lesions of the AP or NTS are also able to block glucagon anorexigenic action [26]. There is also considerable experimental evidence indicating that the hypothalamus plays a main role in the action of glucagon on food intake. For example, the administration of glucagon at low doses into the third ventricle of rats suppressed feeding with a potency greater than 1000 times than that of peripheral administration [27]. Also, glucagon is able to cross the blood–brain barrier [28], and multiple regions in the brain, including the hypothalamus, have demonstrated significant binding to glucagon via GCGR [29]. Indeed, mechanistic studies recently deciphered the neuronal pathway that regulates the anorexigenic actions of glucagon [20]. This process takes place in the hypothalamic arcuate nucleus (ARC) via the GCGR and includes the protein kinase A/ Ca^2+−^calmodulin-dependent protein kinase kinase β/AMP-activated protein kinase/Agouti related protein (PKA/CaMKKβ/AMPK/AgRP) signaling pathway [20] (Figure 1). An important aspect of this study was that, in diet-induced obese (DIO) rats, the amount of CaMKKβ and pAMPK in the hypothalamus and the food intake did not change after glucagon treatment. Nevertheless, the anorectic action of central glucagon in obese rats could be restored by using virogenetic tools to inhibit CaMKKβ activity within the ARC, suggesting that DIO-induced resistance to the anorexic function of glucagon can be explained by deficient CaMKKβ signaling [20]. These results suggest that the diet-induced resistance to the anorexigenic action of glucagon might contribute to the development of obesity.

It is important to note that not all studies found an effect of glucagon on food intake. For example, glucagon administration for more than six months does not affect feeding in a model with impaired leptin signaling [30]. However, is important to note that in this study glucagon was not able to change other relevant metabolic parameters such as glucose, insulin, ketone bodies, or circulating lipids, suggesting that the differences observed with other studies (see next sections) may be explained by the singularity of the model used and in the different doses and time of administration of glucagon. Apart from this, since obesity is mainly induced by high caloric consumption and that glucagon induces satiety, it is conceivable that this hormone could have an impact on energy metabolism and body weight control.

## 3. Glucagon on Body Weight, Energy Expenditure and Thermogenesis

### 3.1. Glucagon on Body Weight

In addition to the modulating role of glucagon on satiety, glucagon also controls body weight by promoting weight loss in physiological and pathological states in humans and rodents [18,31,32]. To evaluate the effect of glucagon signaling on body weight, loss of function models for the study of glucagon receptor were developed. In this regard, GCGR (–/–) mice have normal body weight when fed normal chow [33] but exhibit lower blood glucose, hyperglucagonemia, and pancreatic α-cell hyperplasia [33]. However, GCGR (–/–) mice are resistant to DIO with decreased body weight (~30%), reduced food intake, improved glucose tolerance, and less gastric emptying [34]. According to the lean phenotype, GCGR (–/–) mice on high fat diet (HFD) exhibit significantly lower white adipose tissue (WAT) and brown adipose tissue (BAT) mass than wild type (WT) mice. Consistent with reduced adipose tissue, plasma leptin levels were also significantly lower in those mice [34]. As we commented on before, another study evaluated the effect of glucagon on body weight in Zucker rats, a genetic model of obesity that exhibit decreased release of glucagon from pancreatic islets, as a consequence of leptin receptor mutation [35]. In these rats, glucagon caused a marked reduction of body weight in a food intake independent manner [30]. These data suggest that the glucagon action on body weight is complex and involve food dependent and independent mechanisms.

### 3.2. Glucagon on Energy Expenditure

In a seminal study in 1957 by Davidson et al., it was reported that glucagon injected subcutaneously increased oxygen consumption in rats [36]. These effects were also found in humans, indicating that hyperglucagonemia during insulin deficiency promotes an increase in energy expenditure (EE), which may contribute to the catabolic state in many conditions [37]. Moreover, this notion is supported by reports which show that rats treated with glucagon gain less weight and less fat compared to pair-fed rats [38]. Accordingly, it was also noted that long-term administration of glucagon reduced body weight (~20%) with no change in feeding, suggesting that the reduced body weight is related to an increase in EE [30]. Recent studies with glucagon agonists provide more mechanistic data on this issue. In this sense, Habegger et al. showed that the subcutaneous administration of the glucagon agonist IUB288 decreased body weight and fat mass in DIO mice by stimulating EE and locomotor activity through the action of fibroblast growth factor 21 (FGF21) [39]. A follow up study determined that the farnesoid X receptor signaling in the liver in concert with FGF21, were responsible for this stimulation of EE and fatty oxidation [40]. Therefore, it seems clear that glucagon impacts body weight through feeding-independent mechanisms, most probably by stimulating EE and fatty oxidation (Figure 2).

### 3.3. Glucagon on Thermogenesis

In addition to enhance metabolic rate in rats, measured by significant elevation of oxygen consumption, administration of glucagon is also able to increase BAT mass and to stimulate BAT thermogenic capacity *in vivo* and *in vitro* [41]. It is important to note that although most of the studies of glucagon on BAT activity were made under supraphysiological doses, another study also recapitulated the thermogenic aspects of glucagon by using physiological doses [42]. Accordingly, after cold exposure the levels of circulating glucagon are significantly elevated in humans and rats [43], suggesting that glucagon participates in the cold-induced thermogenesis. To elucidate this aspect, Kinoshita et al. used mice deficient in proglucagon-derived peptides (GCGKO mice) [44] and showed that GCGKO mice displayed cold intolerance and impaired thermogenesis after cold stimulus that was partially ameliorated by glucagon supplementation [44]. In addition, glucagon administration also restored uncoupling protein 1 (UCP1) expression in BAT and significantly increased circulating FGF21 in these mice [44], suggesting that glucagon modulates BAT function through FGF21 signaling. Accordingly, a recent study also reported that glucagon increased EE in normal mice but this effect was partially blocked in FGF21 (–/–) [45]. However, in the same study the activation of EE by glucagon was also observed in UCP1 (–/–) mice and GCGR^BAT^ conditional knockout mice, suggesting that glucagon increases EE through UCP1 independent pathways [45]. In humans, Salem et al. described the effects of glucagon infusion on EE through a BAT-independent mechanism [46]. Therefore, the importance of BAT in the thermogenic action of glucagon is an issue that remains controversial. Nevertheless, pharmacological studies in rodents provide evidence towards a CNS-sympathetic nervous system-dependent action of glucagon on BAT thermogenesis [47,48,49]. In fact, the inhibition with the β-adrenergic blocker propanol [49] or surgically denervation of BAT [47] blunted the thermogenic effects of exogenous glucagon in rodents. However, more studies are needed to evaluate the contribution of glucagon on BAT thermogenesis in humans, but it seems clear that beyond inducing satiety, glucagon decreases body weight by activating EE and thermogenesis (Figure 2). Due to the fact that glucagon improves body weight by the simultaneous activation of the abovementioned processes and that WAT constitute around 20% of total body composition, it is reasonable to assume that this hormone also modulates WAT lipid metabolism (Figure 2).

## 4. The Effect of Glucagon on Lipid Metabolism

Studies in the early 1960s showed that glucagon administration had a potent hypolipidemic effect [50,51,52]. Glucagon, significantly decreased cholesterol levels and plasma total lipids after its intravenous administration [53]. Glucagon also decreased triglycerides (TG) production [54] and circulating TG, very low-density lipoprotein, and cholesterol in hyperlipidemic and non-hyperlipidemic rats [55]. Thus, it may contribute to the hypolipidemic effect by an inhibition of amino acid incorporation into hepatic lipoprotein [55]. However, these studies did not find lower levels of lipids in liver or erythrocytes. Accordingly, another study demonstrated that subcutaneously injections of glucagon during 21 days, decreased cholesterol and triacylglycerol levels by 40 and 70% in plasma but not in the liver, suggesting a transformation of cholesterol into bile acids and higher urinary secretion of cholesterol by chronic administration of this peptide [56]. The effect of chronic treatment of glucagon on lipoprotein composition demonstrated it to be a potent hypolipidemic agent affecting mainly the apo-E-rich lipoproteins [57,58]. The effect of glucagon on lipid metabolism is mediated by inhibition of lipogenesis and stimulation of lipolysis. Glucagon promotes lipolysis by enhancing the activity of hormone-sensitive lipase, the key enzyme that mobilizes the stored fats by TG hydrolysis in adipocytes [59]. Another route by which glucagon regulates lipid metabolism involves the up-regulation of other lipolytic hormones such as growth hormone [60], cortisol [61], or epinephrine [62]. It was also reported that glucagon receptor agonism improved body fat and plasma cholesterol via FGF21 in rodents and human [39].

On the other hand, studies in GCGR (–/–) mice generated controversial data on hypolipidemic effect of glucagon. Connarello et al. demonstrated that GCGR (–/–) mice were resistant to HFD and the development of hepatic steatosis, probably due to elevated circulating GLP-1 in these mice [34]. However, Longet et al. reported increased hepatic TG secretion following fasting and enhanced susceptibility to hepatosteatosis following exposure to a HFD in the same strain, suggesting that disruption of GCGR signaling impairs the control of fatty acid oxidation during fasting [63]. These studies showed contradictory results despite identical background in both models and similar diets, which demonstrates that further research is needed to elucidate the contribution of glucagon on hepatic steatosis.

Glucagon also affects lipid metabolism via a direct effect on hepatic ketogenesis [64,65]. Glucagon activate fatty acids oxidation under conditions of limited energy supply, such as fasting or uncontrolled type 1 diabetes, resulting in ketogenesis [66]. The lipolytic effect of glucagon provides a constant supply of non-esterified fatty acids to the liver, which leads to maintained fatty acid oxidation and ketone bodies production [67,68]. The role of glucagon in the regulation of ketogenesis was reported by Gerich et al., who showed that suppression of glucagon, caused by the treatment with somatostatin, prevented the development of ketoacidosis in patients with type 1 diabetes mellitus after insulin withdrawal [69]. However, the normal ketone response to physiological elevations of glucagon in healthy humans involves an increase in circulating insulin and decreased ketone bodies production [70], suggesting that the activity of glucagon on ketogenesis can be modulated by the simultaneous actions of other hormones or substrates like insulin [70]. Accordingly, in perfused liver, exposure to insulin inhibits ketone body production [70,71]. These effects were also demonstrated in human ketoacidosis by administration of insulin that rapidly reduced plasma ketone [72].

In conclusion, glucagon acts directly on the main fat storage tissues to control lipid metabolism. However, is important to note that while glucagon induces a hypolipidemic action in WAT, and this is not a matter of discussion, the actions of glucagon on hepatic steatosis and ketogenesis remain controversial, and in the case of the ketogenesis could be indirectly dependent of other hormones such as insulin. Collectively all of these studies have positioned glucagon as a key hormone in the control of lipid metabolism as a mechanism to reduce body weight.

## 5. Novel Approaches against Obesity and Diabetes Targeting the Glucagon Receptor

The pleiotropic actions of glucagon on energy balance have, over the last years, made it an attractive molecule for the treatment of diabetes and obesity. The most promising approach to use glucagon as a therapy against obesity comes from its combination with other hormones. Specifically, the design of single-molecule peptides that integrate the complementary actions of multiple endogenous metabolically-related hormones, known as unimolecular polypharmacy, has become successful (Figure 3).

Remarkably, this strategy has been shown to be more effective than approaches that combine single hormones as separate molecules [73,74]. In this last section, we will focus in the unimolecular polypharmacy of GCGR-based drugs [75]. This strategy presents tremendous efficacy in preclinical animal models and has showed some promising results in preliminary clinical trials. Although this aspect has been extensively reviewed before [75,76,77], we will summarize some of the main studies using these compounds targeting GCGR.

### Dual and Triple Agonists Approach Targeting GCGR

A pioneer compound simultaneously activated the GCGR and the glucagon-like peptide-1 receptor (GLP1R), which promoted weight loss and decreased adiposity while stimulating lipolysis and enhance glucose tolerance and leptin sensitivity in DIO mice [78,79].

Another dual agonist that was based on the sequence of the hormone OXM, a gastrointestinal peptide that is able to act through GLP-1R and GCGR with balanced potency, displayed remarkable effects on satiety, body weight and glucose levels [80]. A third class of dual agonist for these receptors was based in the sequence of the native GLP-1 agonist Exendin-4 peptide and a modified glucagon sequence [81]. This chimera showed impressive metabolic actions in DIO mice and indeed doubled the body weight loss induced by the approved drugs for the treatment of obesity [81]. GLP-1/Glucagon co-agonist, in addition to decrease body weight in rodents and cynomolgus monkeys [82], also showed beneficial effects in humans. One such co-agonist, known with the name of MEDI0382, decreased hyperglycemia and body weight in a Phase II clinical trial in diabetic patients [83]. A recent study with another dual agonist of GLP-1/GCGR, known as SAR425899, also reported similar effects in patients with T2DM [84]. 

Taken together, available evidence obtained with different molecules acting as dual agonists of GLP-1R and GCGR have fulfilled the original expectations of obtaining a much more efficacious treatment than treatment with single agonists. Due to the success with dual agonists in preclinical and clinical studies, this novel strategy has advanced towards the development of single molecules targeting three target receptors at once. Although several early tri-agonists were proved in preclinical studies with suboptimal effects due to an unbalanced agonism [85,86,87], a novel class of a tri-agonist peptides, targeting GLP-1/Glucagon/Glucose Insulinotropic Peptide (GIP) receptors, was developed with promising results [88]. This tri-agonist peptide exhibited a high synergistic activity at each of the three target receptors and greater potency than the native ligands separately [88]. In this study, a low dose of the tri-agonist drastically reduced body weight and adiposity and improved glucose tolerance. Remarkable, this tri-agonist displays higher efficiency compared to other approved drugs, and indeed improves additional parameters such as lipid metabolism and hepatic steatosis in different rodent models of obesity [88,89]. Noticeable, the results obtained showed that this molecule has similar effectiveness on both genders. All these facts implicate that this tri-agonist exhibits high potential translational value.

In summary, this promising strategy showed impressive results on metabolic profile in preclinical animal models targeting GCGR (Figure 3), and due to the superior efficacy compared to currently approved drugs, most of them are now undergoing clinical trials. These studies highlight this strategy as a new attractive path for the managing of metabolic diseases.

## 6. Conclusions

In sum, we have reviewed studies of glucagon action focusing on the energy balance regulation beyond glucose control. Moreover, we summarized some of therapeutic efforts directed at manipulating GCGR signaling for the treatment of metabolic diseases. Glucagon promotes satiety when it is sensed in the hepatic-portal vein through the vagus nerve and further activation of CNS. However, glucagon also promotes weight loss by activation of EE, thermogenesis and fatty oxidation, suggesting that this hormone impacts on body weight through feeding-dependent and independents mechanisms (Figure 2). In line with these glucagon actions leading to reduced body weight and fat loss, the possible involvement of this hormone in other metabolic disease related to obesity such as hepatic steatosis deserves further investigation.

Despite that the physical–chemical properties of glucagon were for long a significant barrier for chronic studies due to the poor solubility in physiological buffers, new strategies using agonists targeting GCGR have become successful approaches to evaluate the multifaceted nature of glucagon signaling in energy balance and metabolic syndrome. Likewise, although substantial progress has been made in the last decades in the understanding of the molecular pathways and physiological systems that governed energy balance, no successful pharmacological treatments for obesity have been developed yet. Currently the most effective method to improve body weight and glucose levels is bariatric surgery. However, bariatric surgery is restricted to patients with morbid obesity due to its perioperative risks by which a pharmacological option to treat obesity is largely desired. In this scenario, we believe that therapeutic approaches based on unimolecular polypharmacy, such as GCGR-based drugs, may represent a relevant alternative to surgical procedures in the treatment of metabolic diseases.

## Figures and Tables

**Figure 1 ijms-20-03905-f001:**
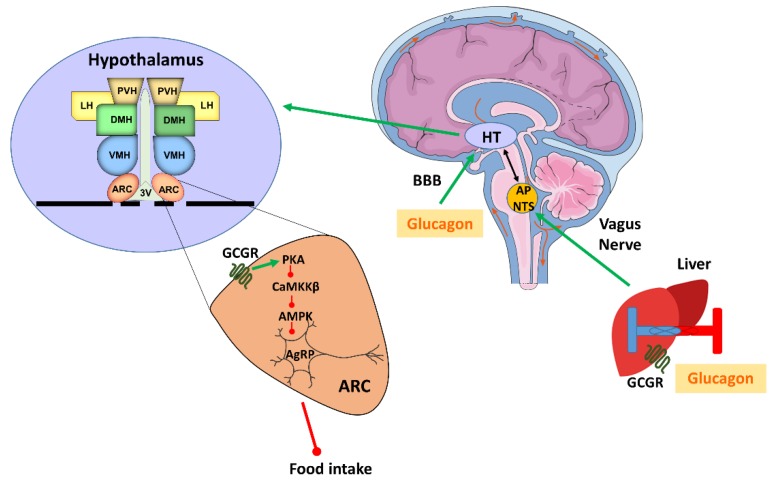
Glucagon control on food intake. Diagram of the neuronal pathway that regulates the anorexigenic action of glucagon. Green arrows indicate an increase or activation, while red arrows indicate a decrease or inhibition. AgRP: Agouti related peptide; AMPK: Adenosine monophosphate -activated protein kinase; AP: Area Postrema; ARC: Hypothalamic arcuate nucleus; BBB: Blood brain barrier; CaMKKβ: Ca2+/calmodulin-dependent protein kinase kinase β; DMH: Dorsomedial hypothalamic nucleus; GCGR: Glucagon receptor; HT: Hypothalamus; LH: Lateral hypothalamic area; NTS: Nucleus tractus solitarius; PKA: Protein kinase A; PVN: Paraventricular hypothalamic nucleus; VMH: Ventromedial hypothalamic nucleus; 3V: Third ventricle. Figure made with Servier Medical Art resources.

**Figure 2 ijms-20-03905-f002:**
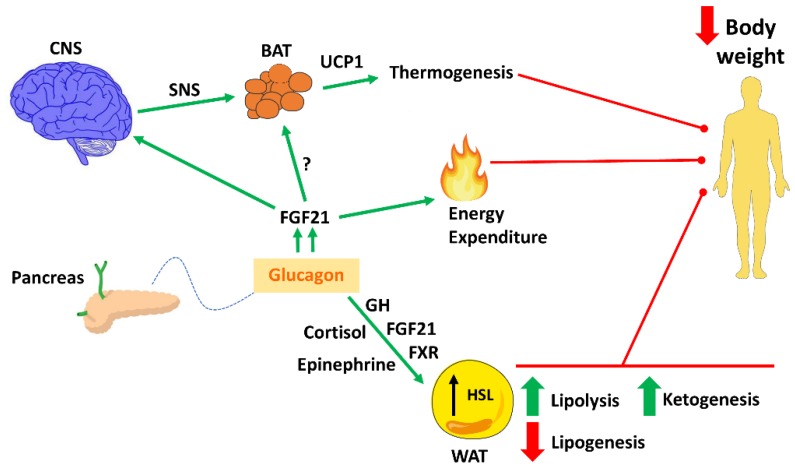
Glucagon on energy balance regulation. Schematic overview of the effects of glucagon on energy balance that finally leads to loss of body weight. Green arrows indicate an increase or improvement, while red arrows indicate a decrease or inhibition. BAT: Brown adipose tissue; CNS: Central nervous system; GH: Growth hormone; HSL: Hormone sensitive lipase; FGF21: Fibroblast growth factor 21; FXR: Farnesoid X receptor; SNS: Sympathetic nervous system; UCP-1: Uncoupling protein 1; WAT: White adipose tissue. Figure made with Servier Medical Art resources.

**Figure 3 ijms-20-03905-f003:**
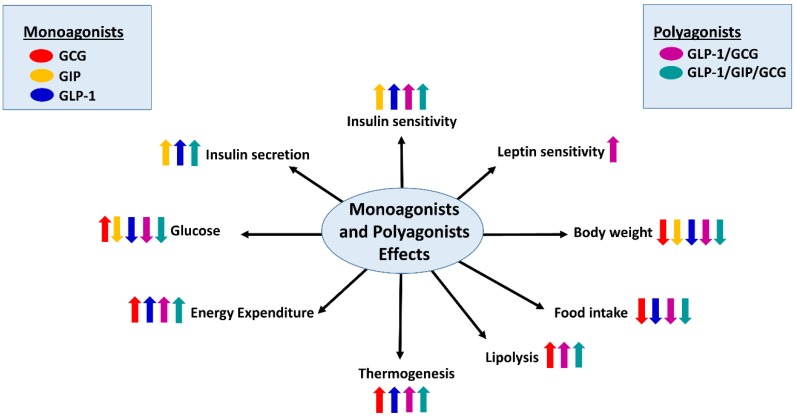
Unimolecular polypharmacy targeting the glucagon receptor: Schematic overview of the effects of polyagonists of glucagon receptor on energy balance regulation. GCG [Glucagon (red)], GIP [glucose insulinotropic peptide (orange)], GLP-1 [Glucagon like peptide-1 (blue)], GLP-1/GCG (purple), and GLP-1/GIP/GCG (green) (right and left panels). Arrows up indicate increase, while arrows down indicate decrease. Figure made with Servier Medical Art resources.

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
