# Peer review of "Glucagon Control on Food Intake and Energy Balance"

_ijms, 2019, doi:10.3390/ijms20163905_

Round 1
Reviewer 1 Report
In this review, the authors aimed to cover metabolic actions caused by and summarizing some of the studies. Indeed, this is a huge field and in a mini-review, it is almost impossible to cover all the aspects. However, the authors tried to cover specifics of the field. By adding or modifying the context in a constructive manner, this review could provide an excellent impact.
Comments:
1. Title of the review is boring. However, the title does not represent the content of the review. By reading the full review I would say, the title should be – “Glucagon control on food intake and energy balance”.
2. The abstract is pretty broad-minded, not focused. Anyone reading this review will expect some take-home messages, which is missing. Also, similar to title, abstract talks about energy balance and metabolism but the title does not seem to connect.
3. Structure of the introduction should come up with three different section (or paragraphs). Background, glucagon peptides and author’s approach in this review.
4. Line # 47-50
“Most of these proglucagon-derived peptides show very specific effects on glucose and energy balance, although some of them exhibit unclear functional roles”
- Authors should provide the reference of actual research instead of Drucker D. et al’s perspective.
· Line # 69-70
“Glucagon exerts effects on metabolism beyond the regulation of glucose metabolism, including modulation of satiety as demonstrated both in humans [14,15] and in rats [16]”
- Authors should provide updated and more rodent research citation in this kind of statements.
· The second sentence under “Glucagon on food intake” was very confusing as it comes out suddenly, I think the title of this section would be better suited to say “Glucagon effect on satiety and suppression”.
· I think their figure would better include just the brain area of ARC because the authors discuss that a lot and they don’t bring up the location of AgRP so it was pretty confusing.
· One of the major complaints of this review is the schematic diagram. Unfortunately, this diagram represents neither signalling pathway, not energy balance or tissue distributions. I would strongly suggest adding a minimum of two distinct figures where one should explain the tissue distribution or release along with energy balance and thermogenesis (which could be done by modifying this current figure); the other one should explain pathways of neurological control. Also, I did not understand what the authors are trying to show by the current figure?
· Line # 96-99
“This process takes place in the hypothalamic arcuate nucleus (ARC) via the GCGR and includes the protein kinase A/ Ca2+-calmodulin-dependent protein kinase kinase β /AMP activated protein kinase (PKA/CaMKKβ/AMPK) signalling pathway”
- Indicates figure 1. How this represents in figure1?
· Glucagon on Body weight, section is an open ending section. Summary of researches make sense, but what the authors are trying to say?
· Line# 124-128
“Another interesting study evaluated the effect of glucagon on body weight in Zucker rats, a genetic model of obesity that exhibit decreased release of glucagon from pancreatic islets, as a consequence of leptin receptor mutation. In these rats, glucagon caused a marked reduction of body weight in a food intake independent manner”
- In what aspect this study become interesting? what the authors are trying to say?
· Bodyweight/Energy expenditure/Thermogenesis should be under the same umbrella with three different sections. Also referring to a new suggested figure would make this review more interesting.
· Energy expenditure and thermogenesis are in the same section, however, should be categorized as mentioned in the previous comment
· Lipid metabolism section is quite structured compared to all other section in the review. However, the authors should state their own comments and judgement based on this section. Other than just saying – “Collectively, all of these studies have positioned glucagon as a key hormone in the control lipid metabolism”
· There are a number of statements where authors specifically mentioned about novel or major studies. How do authors decide a category of some study? For example
Line# 226:
“we will summarize the main studies using these compounds targeting GCGR.”
· In conclusion, the authors suddenly talk about surgical benefit. Should be addressed clearly with additional reference what they are trying to say.
As a suggestion review should improve:
· Title, abstract and conclusion should be in line with the context.
· Figures should be descriptive
· Categorize as
- effects on food intake or CNS control, along with authors specific directions and how it can affect energy metabolism
leads to …….
- effects on Body weight, energy and thermogenesis, along with authors specific directions and how it has control on lipid metabolism
leads to …….
- effects on lipid metabolism, along with authors specific directions
- Novel approaches.
- Conclusion
· Justify why specific research has been picked to discuss.
Author Response
Dear Editors, Dear Dr. Müller and Pr. Habegger,
Thank you for your mail about our review manuscript entitled “Glucagon control on food intake and energy balance” to International Journal of Molecular Sciences.
We thank the Referees for their thorough analyses and thoughtful suggestions. We have taken all their comments carefully into account, addressed their suggestions and extensively revised our manuscript. The point by point responses are described in the attached “Responses to Referees comments”.
We believe that as a result the manuscript has been substantially improved.
We hope that the revised manuscript will now been found suitable for publication in International Journal of Molecular Sciences.
We thank again the Reviewers for their work and comments.
Kind regards,
Sincerely yours,
Mar Quiñones, PhD
In this review, the authors aimed to cover metabolic actions caused by and summarizing some of the studies. Indeed, this is a huge field and in a mini-review, it is almost impossible to cover all the aspects. However, the authors tried to cover specifics of the field. By adding or modifying the context in a constructive manner, this review could provide an excellent impact.
We thank the Reviewer for the time dedicated to revise the manuscript and for his/her so precise comments.
Comments:
1. Title of the review is boring. However, the title does not represent the content of the review. By reading the full review I would say, the title should be – “Glucagon control on food intake and energy balance”.
RESPONSE: We agree with this comment, and we have changed the title of the manuscript following his/her suggestion. The new title is: “Glucagon control on food intake and energy balance”.
2. The abstract is pretty broad-minded, not focused. Anyone reading this review will expect some take-home messages, which is missing. Also, similar to title, abstract talks about energy balance and metabolism but the title does not seem to connect.
RESPONSE: We thank the reviewer for this comment and we have now rewritten the abstract following his/her recommendation. We believe that after this suggestion the abstract is more focused and is in line with the new title.
Revised text, section abstract:
Glucagon exerts pleiotropic actions on energy balance and has emerged as an attractive target for the treatment of diabetes and obesity in the last years. Glucagon reduces body weight and adiposity by suppression of appetite and by modulation of lipid metabolism. Moreover, this hormone promotes weight loss by activation of energy expenditure and thermogenesis. In this review, we cover these metabolic actions elicited by glucagon beyond its canonical regulation of glucose metabolism. In addition, we discuss recent developments of therapeutic approaches in the treatment of obesity and diabetes by dual- and tri-agonist molecules based on combinations of glucagon with other peptides. New strategies using these unimolecular polyagonists targeting the glucagon receptor (GCGR), have become successful approaches to evaluate the multifaceted nature of glucagon signaling in energy balance and metabolic syndrome.
3. Structure of the introduction should come up with three different section (or paragraphs). Background, glucagon peptides and author’s approach in this review.
RESPONSE: We agree with this comment, and we have reorganized the introduction in four different sections. Background, glucagon peptides, glucagon receptor and objectives.
4. Line # 47-50
“Most of these proglucagon-derived peptides show very specific effects on glucose and energy balance, although some of them exhibit unclear functional roles”
Authors should provide the reference of actual research instead of Drucker D. et al’s perspective.
RESPONSE: We apologize for this error, which is corrected in the revised manuscript, and we have included the original articles in the paragraph added below:
Revised text, section 1, new line 47:
Most of these proglucagon-derived peptides show very specific effects on glucose and energy balance, although some of them exhibit unclear functional roles [6-10].
5. Line # 69-70
“Glucagon exerts effects on metabolism beyond the regulation of glucose metabolism, including modulation of satiety as demonstrated both in humans [14,15] and in rats [16]”
Authors should provide updated and more rodent research citation in this kind of statements.
RESPONSE: According to the reviewer, in the revised manuscript we have now included more contemporary original articles in this paragraph.
The new references are: (PMID:26909312; Ref 20 and PMID:11557638; Ref 21)
Revised text, section 2, new line 70:
Glucagon exerts effects on metabolism beyond the regulation of glucose metabolism, including modulation of satiety as demonstrated both in humans [18,19] and in rats [20,21].
6. The second sentence under “Glucagon on food intake” was very confusing as it comes out suddenly, I think the title of this section would be better suited to say “Glucagon effect on satiety and suppression”.
RESPONSE: We agree with this comment, and we have changed the title of the section and revised the manuscript following his/her suggestion.
Revised text, section 2, new line 69/71:
2. The effect of glucagon on satiety and appetite suppression
The effect of glucagon on satiety was shown to be mainly due to the inhibition of meal size via Liver-Brain axis
7. I think their figure would better include just the brain area of ARC because the authors discuss that a lot and they don’t bring up the location of AgRP so it was pretty confusing.
8. One of the major complaints of this review is the schematic diagram. Unfortunately, this diagram represents neither signalling pathway, not energy balance or tissue distributions. I would strongly suggest adding a minimum of two distinct figures where one should explain the tissue distribution or release along with energy balance and thermogenesis (which could be done by modifying this current figure); the other one should explain pathways of neurological control. Also, I did not understand what the authors are trying to show by the current figure?
RESPONSE to points 7 and 8: Due to the referee 3 has the same concern about the figures than referee 1, we redraw the figure 1 and we added 2 more figures to improve the understanding of the different sections. We found this suggestion very valuable and now believe that the text is easier to follow with the new diagrams.

Reviewer 2 Report
p.p1 {margin: 0.0px 0.0px 0.0px 0.0px; text-align: justify; font: 11.0px Helvetica; -webkit-text-stroke: #000000} p.p2 {margin: 0.0px 0.0px 0.0px 0.0px; text-align: justify; font: 11.0px Helvetica; -webkit-text-stroke: #000000; min-height: 13.0px} span.s1 {font-kerning: none}Al-Massadi et al presented a review on the glucagon action on energy balance. It is a very relevant subject regarding glucagon metabolic actions and novel approaches targeting the glucagon receptor.
MAJOR CHANGES
The title needs to better reflect the full content of the review and not only the first two chapters.
The references would need to be revised as from the initial chapters most of the references are from the 1980-90’s or even earlier. For example, in Chapter 2, from 13 references there are only 2 after 1984. Aren’t more recent studies available from glucagon administration or its receptor modulation, which have evaluated feeding behaviour? In this same chapter, almost a third of the text is of auto-citation, which should be, specially when so explicit, avoided.
There is a serious inconsistency in the data referring food intake. Although the authors conclude that glucagon exerts a anorexigenic action (Chapter 2), they later refer another study where “no change in feeding” was observed (Chapter 4, lines 135-137). These negative results, regarding the previous statement, should be included in Chapter 2 and discussed.
MINOR CHANGES
Polipharmacy should not be included as a significant keyword.
The article should not be auto-referred as “mini-review” (line 63).
“Several studies” is stated (line 73) but then only a reference, and from 1984, is indicated.
The authors should avoid to use adjectives such as “interesting study”; aren’t they all?
What does “to trigger BAT mass” (line 146) means?
Referring editing, there’re a few extra spaces that need to be deleted (lines 71, 126, 211, 230) and not-highlighted (160, 212, 239).
Chapter 7 should be included in the previous chapter, since the authors say “in this last section (…) we will summarize” (line 223), but then they include a totally new section.
Author Response
Dear Editors, Dear Dr. Müller and Pr. Habegger,
Thank you for your mail about our review manuscript entitled “Glucagon control on food intake and energy balance” to International Journal of Molecular Sciences.
We thank the Referees for their thorough analyses and thoughtful suggestions. We have taken all their comments carefully into account, addressed their suggestions and extensively revised our manuscript. The point by point responses are described in the attached “Responses to Referees comments”.
We believe that as a result the manuscript has been substantially improved.
We hope that the revised manuscript will now been found suitable for publication in International Journal of Molecular Sciences.
We thank again the Reviewers for their work and comments.
Kind regards,
Sincerely yours,
Mar Quiñones, PhD
MAJOR CHANGES
1. The title needs to better reflect the full content of the review and not only the first two chapters.
RESPONSE: We agree with this comment, that was raised also by other referees. We have changed the title of the manuscript following his/her suggestion. The new title is: “Glucagon control on food intake and energy balance”.
2. The references would need to be revised as from the initial chapters most of the references are from the 1980-90’s or even earlier. For example, in Chapter 2, from 13 references there are only 2 after 1984. Aren’t more recent studies available from glucagon administration or its receptor modulation, which have evaluated feeding behaviour?
RESPONSE: Is important to note that although several studies over the past five decades have reported glucagon-stimulated body weight loss and suppression of appetite in both humans and rodents, this phenomenon received limited attention until the group of Matthias Tschӧp and Richard Di Marchi between others recently reported potent body weight–lowering effects of novel multi-agonists based in unimolecular polypharmacy. As a proof of fact Glucagon has 45587 entries and glucagon + food intake 2921 (together with glucagon like peptide 1). Anyway, we add new references that are more contemporary in this chapter when suitable: (PMID:1858943, PMID: 7943304, PMID:920200, PMID:8430871, PMID:11557638).
3. In this same chapter, almost a third of the text is of auto-citation, which should be, specially when so explicit, avoided.
RESPONSE: We really apologize for this, now we have corrected that issue by reduced the length of the paragraph where we talk about our work in the revised manuscript.
Revised text, section 2, line 107:
More specifically, it has been found that the central injection of glucagon increases hypothalamic levels of cyclic AMP response element binding in its phosphorylated (pCREB) form, indicating an activation of PKA, and decreases hypothalamic levels of CaMKKβ and its downstream targets pAMPK and the phosphorylated form of Acetyl Coenzima A carboxylase. Accordingly, the blockade of central PKA or the activation of AMPK by genetic means, specifically in the ARC, were sufficient to blunt the anorexigenic action of glucagon [25]. In addition, icv administration of glucagon induces changes in agouti-related peptide (AgRP) expression and GGCR co-localizes with AgRP neurons [26], suggesting an important role of AgRP neurons in the anorectic action of glucagon.
4. There is a serious inconsistency in the data referring food intake. Although the authors conclude that glucagon exerts a anorexigenic action (Chapter 2), they later refer another study where “no change in feeding” was observed (Chapter 4, lines 135-137). These negative results, regarding the previous statement, should be included in Chapter 2 and discussed.
RESPONSE: We agree with this comment and we have added one paragraph at the end of the section 2.
Revised text, section 2, line 122:
It is important to note that not all studies found an effect of glucagon on food intake. For example, glucagon administration for more than 6 months does not affect feeding in a model with impaired leptin signaling [30]. However, is important to note that in this study glucagon was not able to change other relevant metabolic parameters such as glucose, insulin, ketone bodies or circulating lipids, suggesting that the differences observed with other studies (see next sections) may be explained by the complexity/singularity of the model used and in the different doses and time of administration of glucagon. Besides this, since obesity is mainly induced by high caloric consumption and that glucagon induces satiety, is conceivable that this hormone could have impact on energy metabolism and body weight control
MINOR CHANGES
1. Polipharmacy should not be included as a significant keyword.
The article should not be auto-referred as “mini-review” (line 63).
“Several studies” is stated (line 73) but then only a reference, and from 1984, is indicated.
The authors should avoid to use adjectives such as “interesting study”; aren’t they all?
RESPONSE: We apologize for these errors, which are corrected in the revised manuscript.
2. What does “to trigger BAT mass” (line 146) means?
RESPONSE: We apologize for the misunderstanding. In the referred article have been shown that glucagon increases the weight of the BAT in addition to its activity. We try to better explain this fact as you can see below.
Revised text, section 3.3, line 173:
In addition to enhance metabolic rate in rats, measured by significant elevation of oxygen consumption, administration of glucagon is also able to increase BAT mass and to stimulate BAT thermogenic capacity in vivo and in vitro [41].
3. Referring editing, there’re a few extra spaces that need to be deleted (lines 71, 126, 211, 230) and not-highlighted (160, 212, 239).
RESPONSE: We apologize for these errors, which are corrected in the revised manuscript.
4. Chapter 7 should be included in the previous chapter, since the authors say “in this last section (…) we will summarize” (line 223), but then they include a totally new section.
RESPONSE: We totally agree with this comment and we have included the old chapter 7 in the new chapter 5.

Reviewer 3 Report
This is a well written review focussing on the effects of glucagon on satiety, lipid metabolism, energy homeostasis and body weight regulation. In addition the review describes the advances in developing new dual and tri-agonist drugs based on glucaogn and glucagon like peptides for treating obesity and diabetes.
The following grammatical errors were found and these need to be corrected:
Page 3 line 102: phosphorylated form of Acetyl Coenzima A carboxylase
Page 4 line 161: EE though a BAT-independent mechanism
Page 5 line 199: lipid metabolism via a direct effects on hepatic ketogenesis
Page 5 line 206: mellitus after insulin retreat
Page 5 line 211: that rapidly reduced plasmas
Page 6 line 238: co-agonists in addition to decrease body weight
Currently the review only has 1 figure and this figure lacks detials in terms of the mechanimsms as to how glucagon regulates all these non glucose effects. This figure is a good summary and should be the final figure which summarises all the other concepts developed in the paper.
My suggestions to improve the review would be to have more figures 1) showing the detailed effects of glucagon on brown adipose tissue, white adipose tissue and central nervous tissue 2) the mechanism of action of the newly developed drugs based on glucagon and GLP-1 for potentially treating obesity and diabetes. The pictorial presentations will complement the nicely written review and make it easier for the non experts in the field to understand the concepts.
Author Response
Dear Editors, Dear Dr. Müller and Pr. Habegger,
Thank you for your mail about our review manuscript entitled “Glucagon control on food intake and energy balance” to International Journal of Molecular Sciences.
We thank the Referees for their thorough analyses and thoughtful suggestions. We have taken all their comments carefully into account, addressed their suggestions and extensively revised our manuscript. The point by point responses are described in the attached “Responses to Referees comments”.
We believe that as a result the manuscript has been substantially improved.
We hope that the revised manuscript will now been found suitable for publication in International Journal of Molecular Sciences.
We thank again the Reviewers for their work and comments.
Kind regards,
Sincerely yours,
Mar Quiñones, PhD
1. This is a well written review focusing on the effects of glucagon on satiety, lipid metabolism, energy homeostasis and body weight regulation. In addition, the review describes the advances in developing new dual and tri-agonist drugs based on glucagon and glucagon like peptides for treating obesity and diabetes.
RESPONSE: We would like to thank the Reviewer for the positive and encouraging comments on our manuscript.
2. The following grammatical errors were found and these need to be corrected:
Page 3 line 102: phosphorylated form of Acetyl Coenzima A carboxylase
Page 4 line 161: EE though a BAT-independent mechanism
Page 5 line 199: lipid metabolism via a direct effects on hepatic ketogenesis
Page 5 line 206: mellitus after insulin retreat
Page 5 line 211: that rapidly reduced plasmas
Page 6 line 238: co-agonists in addition to decrease body weight
RESPONSE: We apologize for these errors, which are corrected in the revised manuscript
3. Currently the review only has 1 figure and this figure lacks detials in terms of the mechanimsms as to how glucagon regulates all these non glucose effects. This figure is a good summary and should be the final figure which summarises all the other concepts developed in the paper.
My suggestions to improve the review would be to have more figures 1) showing the detailed effects of glucagon on brown adipose tissue, white adipose tissue and central nervous tissue 2) the mechanism of action of the newly developed drugs based on glucagon and GLP-1 for potentially treating obesity and diabetes. The pictorial presentations will complement the nicely written review and make it easier for the non experts in the field to understand the concepts.
RESPONSE: Due to other referees has the same concern about the figures, we redraw the figure 1 and we added 2 more figures to improve the understanding of the different sections. We believe that this suggestion is very valuable and now the text is easy to follow with the new diagrams.
Figure 1: Glucagon control on food intake.
Figure 2: Glucagon on energy balance regulation.
Figure 3: Unimolecular polypharmacy targeting the glucagon receptor
Round 2
Reviewer 1 Report
In this review, the authors aimed to cover metabolic actions caused by glucagon and summarizing some of the studies. Indeed, this is a huge field and in a short review, it is almost impossible to cover all the aspects. However, the authors tried to cover the specifics of the fields. In the first version, major comments were related to the structure, take-home messages and figures. The authors have modified all those concerns and added new figures. Review flow nicely and there are loads of take-home messages now.
The authors may want to proofread for minor typos and grammatical mistakes.
Reviewer 2 Report
p.p1 {margin: 0.0px 0.0px 0.0px 0.0px; text-align: justify; font: 11.0px Helvetica; -webkit-text-stroke: #000000} span.s1 {font-kerning: none}Al-Massadi et al review on the glucagon action on energy balance has been updated and the authors have positively addressed all my concerns and suggestions. Still, special attention should be given to final corrections and editing, as in line 74, for example.